# Endogenous Opioid Peptides and Alternatively Spliced Mu Opioid Receptor Seven Transmembrane Carboxyl-Terminal Variants

**DOI:** 10.3390/ijms22073779

**Published:** 2021-04-06

**Authors:** Anna Abrimian, Tamar Kraft, Ying-Xian Pan

**Affiliations:** Department of Anesthesiology, Rutgers New Jersey Medical School, Newark, NJ 07103, USA; aa2279@njms.rutgers.edu (A.A.); ttk19@njms.rutgers.edu (T.K.)

**Keywords:** β-endorphin, dynorphin A, [Met]^5^Enkephalin-Arg^6^-Phe^7^, endormorphins, mu opioid receptor, MOR, OPRM1, alternative splicing, G protein, β-arrestin, biased signaling

## Abstract

There exist three main types of endogenous opioid peptides, enkephalins, dynorphins and β-endorphin, all of which are derived from their precursors. These endogenous opioid peptides act through opioid receptors, including mu opioid receptor (MOR), delta opioid receptor (DOR) and kappa opioid receptor (KOR), and play important roles not only in analgesia, but also many other biological processes such as reward, stress response, feeding and emotion. The MOR gene, OPRM1, undergoes extensive alternative pre-mRNA splicing, generating multiple splice variants or isoforms. One type of these splice variants, the full-length 7 transmembrane (TM) Carboxyl (*C*)-terminal variants, has the same receptor structures but contains different intracellular *C*-terminal tails. The pharmacological functions of several endogenous opioid peptides through the mouse, rat and human OPRM1 7TM *C*-terminal variants have been considerably investigated together with various mu opioid ligands. The current review focuses on the studies of these endogenous opioid peptides and summarizes the results from early pharmacological studies, including receptor binding affinity and G protein activation, and recent studies of β-arrestin2 recruitment and biased signaling, aiming to provide new insights into the mechanisms and functions of endogenous opioid peptides, which are mediated through the OPRM1 7TM *C*-terminal splice variants.

## 1. Introduction

Discovery of the three main types of endogenous opioid peptides, enkephalins, dynorphins and β-endorphin in the 1970s [1,2,3,4] with help by early established opioid receptor binding assays [5,6,7] revolutionized the opioid field and further advanced our understanding of opioid receptor subtypes. Decades of research have revealed that all these endogenous opioid peptides play important roles in many biological systems by acting through opioid receptors. Molecular cloning of the delta opioid receptor (DOR-1) in 1992 [8,9] quickly led to isolate the mu opioid receptor (MOR) [10,11,12,13] and kappa opioid receptor (KOR-1) [14,15,16]. These discoveries not only validated the pharmacologically defined opioid receptor subtypes, but also provided essential tools to investigate the mechanisms and functions of the endogenous opioid peptides. A single-copy gene was identified for each of these receptors. The MOR gene (OPRM1) undergoes extensive alternative pre-mRNA splicing, producing multiple splice variants or receptor isoforms (see reviews: [17,18,19]. Although several splice variants were identified in OPRD1 [20,21] and OPRK1 genes [21,22,23], the extent of the OPRM1 alternative splicing is far larger and more complex than the OPRD1 and OPRK1. Conservation of the OPRM1 alternative splicing from rodent to human also suggests the evolutionary importance of the OPRM1 alternative splicing and resulting splice variants.

The relationships between endogenous opioid peptides and originally cloned opioid receptors, including MOR-1, DOR-1 and KOR-1, have been extensively studied in many different systems. In this review, we mainly focus on the pharmacological functions of several endogenous opioid peptides, including β-endorphin, dynorphin A and [Met]^5^Enkephalin-Arg^6^-Phe^7^, through the Oprm1 full-length seven transmembrane (7TM) carboxyl (*C*-) terminal variants in terms of binding affinity, G protein coupling, β-arrestin2 recruitment and biased signaling. We also include the data from endomorphin-1 and endomorphin-2 despite the fact that their precursors and genes have not been identified.

## 2. The Opioid Receptors and Endogenous Opioid Peptides

The opiates derived from opium have been used for thousands of years. However, the concept of opiate receptors was only proposed several decades ago based on the strict structural requirements needed for opiate activity [24,25,26,27,28,29]. Subsequently, Martin proposed the existence of opioid subtypes in his proposal of receptor dualism [30] and then suggested M and N receptors, which later were referred to mu (morphine) and kappa (ketocyclazocine) receptors, respectively [31]. Soon afterwards, the delta-opioid receptor was proposed as the recognition sites for the enkephalins [32,33,34]. In 1973, three laboratories experimentally demonstrated opioid binding sites in the central nervous system for the first time using various ^3^H-labeled ligands, including ^3^H-naloxone [5], ^3^H-dihydromorphine [6] and ^3^H-etorphine [7]. The high stereospecificity and selectivity of the binding for opiates were consistent with the basis for a receptor [35]. Biochemical and pharmacological studies further confirmed the protein nature of the binding sites by their sensitivity to proteases, including trypsin and chymotrypsin [7,36,37], as well as the reagents targeting sulfhydryl groups [36,37], and their insensitivity to DNase, RNase, neuraminidase and phospholipase C [7,36,37].

The identification of opioid receptor binding sites in the brain quickly let to the quest of their endogenous ligands. The endogenous opioid-like substances in the brain were first disclosed by several labs at a meeting of the Neuroscience Research Program in Boston in 1974 sponsored by the Massachusetts Institute of Technology [38]. Subsequently, Kosterlitz and Hughes were the first to report the sequences of two pentapeptide enkephalins [32]. This was quickly followed by the isolation of two other endogenous opioid peptides, dynorphin and β-endorphin [1,2,3,4,39]. Similar to most neuropeptides, all these peptides are produced through post-translational modifications of their precursors, proenkephalin, prodynorphin and proopiomelanocortin (POMC), by several processing enzymes and peptidases (Figure 1) [40]. Both proenkephalin and prodynorphin generate several opioid peptides, while POMC yields only β-endorphin in addition to some non-opioid peptides such as adrenocorticotropin and α-melanocyte-stimulating hormone.

All the endogenous opioid peptides contain the enkephalin sequence, Tyr-Gly-Gly-Phe-Leu or Tyr-Gly-Gly-Phe-Met, at the *N*-terminus with different *C*-terminal sequences (Table 1). The enkephalins are the endogenous ligands for the delta-opioid receptor (DOR-1). Although dynorphins are considered endogenous agonists for the kappa1-opioid receptor (KOR-1), they bind to the mu-opioid receptor (MOR-1) and DOR-1 with high affinities as well [41,42]. Additionally, β-endorphin is thought to be an endogenous agonist of MOR-1, but has high affinity for DOR-1 [42].

Another group of endogenous opioid peptides are endomorphins, including endomorphin-1 (Tyr-Pro-Trp-Phe-NH2) and endomorphin-2 (Tyr-Pro-Phe-Phe-NH2) [43]. Both endomorphins lack the common enkephalin motif (Try-Gly-Gly-Phe) shared by other opioid peptides. However, they are the ligands highly selective for the mu-opioid receptor (MOR-1). The distribution and function of endomorphins have been extensively studied [44]. However, the precursors for these endomorphins or their genes remain to be identified.

Enkephalins are widely distributed in the central nervous system, such as the striatum, hypothalamus, thalamus, hippocampus, pons, medulla and spinal cord. Dynorphins have similar distributions as enkephalins with a few exceptions. POMC is mainly synthesized in the pituitary gland. POMC mRNA is highly expressed in the hypothalamus and detected in the caudal nucleus tractus solitarius and the commissural nucleus, as well as in peripheral tissues such as testis, gut, kidney, adrenal and skin. Extensive studies showed that all these endogenous opioid peptides play important roles in a variety of biological functions. In addition to analgesia, they can modulate reward, addiction, stress response, emotion and feeding (see reviews: [42,45,46,47,48,49,50,51]). Several transgenic mouse models targeting either the precursors or encoded peptides were generated to study in vivo function of these endogenous opioid peptides [52,53,54,55,56].

## 3. Alternative Splicing of Mu-Opioid Receptor Gene, OPRM1

The mu-opioid receptor has a special place within the opioid receptor family because it mediates the actions of most of the clinically used opioids such as morphine and fentanyl, as well as drugs of abuse such as heroin. The existence of multiple mu-opioid receptors has been long suggested by clinical observations that patients often show different sensitivities towards various mu opioids not only in analgesia, but also in their side-effects including tolerance, dependence, itch, constipation and addiction. Furthermore, incomplete cross tolerance in patients has led to the clinical practice of opioid rotation in which patients who develop tolerance to one mu opioid must use much higher doses of the opioid for pain relief can take back analgesic control by switching to another mu opioid with lower doses. Similar observations were seen in animal models [57,58,59,60,61]. It is difficult to interpret these observations using a single mu receptor mechanism. Early pharmacological studies defined mu_1_ and mu_2_ receptors using in vivo behavioral assays and in vitro opioid receptor binding assays with newly synthesized antagonists including naloxazone and naloxonazine [62,63,64,65,66,67] and also morphine-6β-glucuronide (M6G) receptor [68,69,70,71]. However, genomic characterization of the MOR gene using the MOR cDNA clones and the human genome sequencing project revealed only a single copy of the MOR gene, OPRM1, raising questions about how a single copy of OPRM1 gene reconciles multiple mu-opioid receptors suggested by clinical observations and the pharmacological studies.

One hypothesis to address these questions is that the single copy of the OPRM1 gene creates multiple mu-opioid receptor splice variants or isoforms through alternative pre-mRNA splicing. Driven by this hypothesis, many efforts have been made to isolate alternatively spliced MOR variants in the past decades. We now know that the OPRM1 gene goes through extensive alternative splicing, generating an array of splice variants, which is far more complex than those suggested by the early pharmacological studies (see review: [17,18,19]). The OPRM1 alternative splicing is conserved from rodent to human. Interestingly, only the OPRM1 gene, but no other opioid receptor genes, underwent extensive and conserved alternative splicing, suggesting the evolutionary importance of the OPRM1 gene.

The OPRM1 splice variants can be categorized into three main types [18,19]: (1) the full-length 7 transmembrane (TM) *C*-terminal variants produced by alternative 3′ splicing (Figure 2). These 7TM *C*-terminal variants have identical receptor structures including the *N*-terminus, TM regions, intra-/extra-cellular loops and part of intracellular *C*-terminus, except for their differences at the *C*-terminal tails; (2) the truncated 6TM variants that lack the extracellular *N*-terminus and the first TM, generated by a combination of alternative promoter, exon skipping, alternative 5′ and/or 3′ splicing; (3) the truncated 1TM variants that contain only the extracellular *N*-terminus and the first TM, generated by exon skipping or insertion.

The functional relevance of the full-length 7TM *C*-terminal variants has been indicated by their differences in mu agonist-induced G protein coupling [73,74,75,76,77,78,79], β-arrestin2 recruitment [80,81], internalization [82,83], phosphorylation [82] and post-endocytic sorting [84] when expressed in cell lines. The 7TM *C*-terminal variants were differentially expressed in various brain regions or different inbred mouse strains at the mRNA level [85,86], and at the protein level [87,88]. Dysregulation of these variant mRNAs was observed in the medial prefrontal cortex of human heroin abusers and heroin self-administering rats [89], multiple brain regions of morphine tolerant mice [85], and HIV patients [90,91]. Importantly, in vivo functions of these 7TM *C*-terminal variants were demonstrated in morphine-induced tolerance, dependence and reward using several *C*-terminal truncation mouse models [80]. For example, truncating exon 7-encoded *C*-terminal sequences reduced morphine tolerance and reward without the effect on morphine dependence. Conversely, truncating exon 4-encoded *C*-terminal sequences facilitated morphine tolerance and reduced morphine dependence without the effect on morphine reward. The mouse MOR-1D and human MOR-1Y involved morphine-induced itch (pruritus) [92,93].

The truncated 6TM variants mediated the analgesic actions of a subset of mu opioids including heroin, M6G [94], buprenorphine [95] and a novel class of opioid analgesics such as 3′-iodobenzoyl-6β-naltrexamide (IBNtxA) that are potent against a broad spectrum of pain models without many side-effects associated with traditional opiates [96,97]. The 1TM variants did not bind any opioids. However, the 1TM variants can increase expression of 7TM MOR-1 at the protein level as a molecular chaperon to enhance morphine analgesia [98]. The 6TM variants can also facilitate expression of 7TM MOR-1 at protein level through heterodimerization [99].

## 4. Binding Affinities of Endogenous Opioid Peptides in the Full-Length 7TM *C*-terminal Splice Variants

Soon after each 7TM variant cDNAs were cloned, the cell lines that stably expressed each of the individual 7TM variants in Chinese Hamster Ovary (CHO) and Human embryonic kidney (HEK) 293 cells were established [74,75,76,78,79,82,100] and initially used in opioid receptor binding assays to define their binding profiles. Saturation studies using [^3^H][_D_-Ala^2^,N-MePhe^4^,Gly-ol]-enkephalin (DAMGO), a synthetic opioid peptide and a full mu agonist, as indicated by the K_d_ values at subnanomolar range, suggest that [^3^H]DAMGO has a high affinity to all these 7TM *C*-terminal variants. Competition studies using [^3^H]DAMGO with various opioids, such as morphine, M6G and naloxone, further established their mu selectivity by the fact that all mu opioids competed the binding potently, as indicated by the K_i_ values at subnanomolar range, while delta or kappa drugs failed to compete at the concentration of over 500 nM. These results were not surprising because all these 7TM *C*-terminal variants contain the same binding pocket, which is mainly constituted by the transmembrane domains and extracellular loops. However, several endogenous opioid peptides displayed differential binding affinities among the 7TM *C*-terminal variants. Table 2 summaries the results of the K_i_ values of several endogenous opioid peptides against the mouse, rat and human 7TM *C*-terminal variants from several early studies [73,74,75,76,77,78,79,100]. Although these studies were performed at different times when the variants were isolated, the complied data provides reasonable comparisons regarding the binding affinities of the indicated endogenous opioid peptides among the 7TM *C*-terminal variants because all the competition assays were performed using [^3^H]DAMGO with membranes isolated from the stable cell lines using the same parental CHO cells. The K_i_ values of DAMGO and morphine are also listed for the comparison.

One intriguing observation from Table 2 is that the binding profiles of the endogenous opioid peptides among the 7TM *C*-terminal variants were different from those of DAMGO, morphine, fentanyl and methadone, all of which had similar K_i_ values against various 7TM *C*-terminal variants. For example, DAMGO’s K_i_ values had a range of 0.7–3.3 nM among the mouse 7TM *C*-terminal variants, while fentanyl’s K_i_ values arranged from 1.2–3.3 nM among the mouse variants. However, endogenous opioid peptides, particularly dynorphin A and β-endorphin, displayed versatile K_i_ value ranges toward the 7TM *C*-terminal variants. For example, in the mouse 7TM *C*-terminal variants, dynorphin A had higher affinities in mMOR-1C (5.6 nM) and mMOR-1D (2.2 nM), but showed lower affinities in mMOR-1O (58 nM) and mMOR-1P (103 nM), while it had intermediate K_i_ values in other variants. Similarly, there was a 13-fold difference in the K_i_ values of dynorphin A between hMOR-1X (186.8 nM) and hMOR-1B3 (13.8 nM). Similar scenarios were seen in β-endorphin and endomorphins. β-endorphin competed the binding more potent in mMOR-1D (1.7 nM) than in mMOR-1O (16 nM), an over 9-fold difference. Both endomorphin-1 and endomorphin-2 had higher affinities in mMOR-1C compared to lower affinities in mMOR-1B1. Furthermore, M6G showed moderately different K_i_ values, particularly against the human variants. All the 7TM *C*-terminal variants share the same opioid binding pocket but contain a different intracellular *C*-terminal tail sequence. Why can these *C*-terminal sequences away from the binding pocket modulate the binding affinities of the endogenous opioid peptides, but not DAMGO, morphine, fentanyl and methadone? The crystal structure of the MOR in both agonist and antagonist conformations has been resolved [101,102], providing the fundamental basis of our understanding on structural relationships of ligand-receptor interactions. However, these crystal structures were determined by using the *N*-terminal and *C*-terminal truncated receptor to allow for the establishment of the stabilized crystal structures, offering no information on the role of the *C*-terminal sequences on overall MOR structure. Although future structural determination of various *C*-terminal tails’ role on the ligand binding would give an ideal answer to the question, we speculate two possible mechanisms: (1) the intracellular loops, especially the intracellular loop II and III, can impact G protein coupling or receptor agonist conformation. Potential interactions of the *C*-terminal tail sequences with these intracellular loop regions could differentially modulate the receptor agonist conformation especially for the endogenous opioid peptides; (2) several known proteins such as G proteins and β-arrestins or unknown proteins can associate with the MORs at basal or active states, influencing ligand binding. The *C*-terminal tail sequences could alter the receptor agonist conformation mainly for the endogenous opioid peptides by interacting with these associated proteins.

## 5. G Protein Coupling Induced by Endogenous Opioid Peptides in the Full-Length 7TM *C*-terminal Splice Variants

Intracellular location of the alternative *C*-termini raises apparent questions regarding their roles on mu agonist-induced G protein coupling. [^35^S]GTPγS binding assays have commonly been used for measuring ligand-induced G protein coupling in G protein coupled receptors (GPCRs) [103,104]. Using unhydrolyzable GTPγS nature, [^35^S]GTPγS binding assays provide an accurate and sensitive tool to quantify the total amount of G proteins trapped with receptors, although the assays cannot determine which G proteins are involved. The abilities of various mu ligands, including mu opioids and endogenous opioid peptides, in the stimulation of G protein coupling on different 7TM *C*-terminal variants were extensively studied using the same plasma membranes isolated from the CHO cells stably expressing individual variants that were used for opioid receptor binding assays. Table 3 puts together the data from endogenous opioid peptides, as well as DAMGO and morphine, from several published papers [73,74,75,76,77]. Concentration-response curves for each ligand on individual variants were used to determine the potency, indicated by EC_50_ values, and efficacy, indicated by % maximum stimulation (% Max) that was normalized to that of DAMGO for comparisons.

The results revealed marked differences in [^35^S]GTPγS binding by endogenous opioid peptides in both potency (EC_50_ value) and efficacy (% Max) among 7TM *C*-terminal variants. Different intracellular *C*-terminal tails significantly affected the potency of endogenous opioid peptides, particularly β-endorphin. For example, the EC_50_ values of β-endorphin differed over 28-fold between mMOR-1O with a 30-amino acid (aa) *C*-terminal tail encoded by exon 7 and mMOR-1B2 with a different 23-aa *C*-terminal tail encoded by exon 5b. mMOR-1C has an identical exon 7-encoded 30-aa as mMOR-1O, but contains an additional 22-aa *C*-terminal sequence encoded by exons 8/9 (see the sequences in Figure 2). Interestingly, the 22-aa sequences in mMOR-1C increased β-endorphin’s EC_50_ value by 20-fold over mMOR-1O. β-endorphin was more potent in hMOR-1 with a 12-aa tail encoded by exon 4 (4 nM) than in hMOR-1B2 with a totally different 9-aa tail encoded by exon 5b (73 nM). Furthermore, dynorphin A EC_50_ values varied over 6-fold between mMOR-1 and mMOR-1B2 and over 7-fold between hMOR-1A and hMOR-1B2. One intriguing observation is that there was no correlation between the potency (EC_50_ value) of mu agonists, including endogenous opioid peptides, to activate [^35^S]GTPγS binding and their binding affinity (K_i_ value) (Figure 3). For example, [Met]^5^Enkephalin-Arg^6^-Phe^7^ had a wide range of EC_50_ values (29–170 nM) among the mouse 7TM *C*-terminal variants, despite that its K_i_ values from the binding assays were very similar (Table 2). The binding affinity of β-endorphin in mMOR-1O was over 9-fold lower than in mMOR-1D. Contrastingly, β-endorphin was more potent in stimulating [^35^S]GTPγS binding in mMOR-1O (EC_50_: 6 nM) than in mMOR-1D (EC_50_: 73 nM). The mismatch between the K_i_ and EC_50_ values suggests that different *C*-terminal sequences can impact on the potency of endogenous opioid peptides to stimulate [^35^S]GTPγS binding independent of their binding affinity.

The relationship between the K_i_ and EC_50_ values can also be indicated by the EC_50_/K_i_ ratio, which represents both an assessment for the ability of an agonist to stimulate the receptor in [^35^S]GTPγS binding relative to its receptor occupancy or binding affinity, and an indirect indication of intrinsic activity (Table 4). Again, we observed a wide range of the EC_50_/K_i_ ratios among 7TM *C*-terminal variants for endogenous opioid peptides, particularly β-endorphin, consistent with no correlation between the K_i_ and EC_50_ values. What is most striking is that the ratios between mMOR-1O and mMOR-1D differed over 113-fold. Additionally, there was an 18-fold difference of β-endorphin EC_50_/K_i_ ratio between hMOR-1 and hMOR-1Y, and a 10-fold difference between rMOR-1 and rMOR-1D. The EC_50_/K_i_ ratios of dynorphin A varied over 14-fold between mMOR-1 and mMOR-1D and 8-fold between hMOR-1B3 and hMOR-1B4. These results suggest that the *C*-terminal tail sequences have significant impact on the intrinsic activity of mu agonists including endogenous opioid peptides.

The relative efficacy or % maximum stimulation (% Max) of endogenous opioid peptides in stimulation of [^35^S]GTPγS binding varied markedly among the 7TM *C*-terminal variants (Table 3). For example, β-endorphin was a full agonist in mMOR-1D (105%), mMOR-1E (130%) and mMOR-1O (141%), while it became a partial agonist in mMOR-1C (44%) and mMOR-1P (55%). Interestingly, both β-endorphin and dynorphin A were a partial agonist in hMOR-1, hMOR-1B3 and hMOR-1B4, but a full agonist in hMOR-1B2. Similarly, the efficacy of endomorphin-1 and endomorphin-2 differed among the mouse 7TM variants. Just as there was no correlation between the K_i_ and EC_50_ values, there was no correlation between the EC_50_ values and % Max of endogenous opioid peptides among the 7TM *C*-terminal variants.

Together, these results suggest that different intracellular *C*-terminal tails greatly impact Receptor-G protein coupling induced by the endogenous opioid peptides. It should be pointed out that the influence of the *C*-terminal tails on G protein coupling was also observed by most mu agonists such as DAMGO, morphine, fentanyl, and methadone [73,74,75,76,77,78,79,81], in contract to their unchanged binding affinity (Table 2). These results suggest the differential effects of the *C*-terminal tails on ligand binding and G protein activation between endogenous opioid peptides and other mu agonists. The above-mentioned studies were performed using CHO cells and the results may be irrelevant in vivo. However, it is difficult to determine the impact of endogenous opioid peptide-induced Receptor-G protein coupling or their binding affinity on individual 7TM variants in vivo since they co-exist in the brain. It would be interesting to further explore in vivo functional relevance of these differentially expressed 7TM *C*-terminal variants in the Receptor-G protein coupling induced by endogenous opioid peptides using new gene targeting animal models in which only one individual 7TM *C*-terminal variant is expressed. Region-specific, cell-specific, or strain-specific expression of the OPRM1 splice variants including the 7TM *C*-terminal variants were observed at both mRNA [85,86,89] and protein levels [87,88,105] in animals and humans, raising questions whether their roles in mu agonist-induced G protein coupling are region-specific or cell-specific.

## 6. Biased Signaling of Endogenous Opioid Peptides in the Full-Length 7TM *C*-Terminal Splice Variants

Originally, G protein coupled receptors (GPCRs) were defined to signal through interactions with G proteins that transduce their downstream signaling cascades. However, GPCRs have been found to couple non-G protein transducers, such as β-arrestins, to produce G protein-independent signaling, leading to the concept of biased signaling, or biased agonism, or functional selectivity in which different agonists can trigger divergent signaling pathways via the same receptor and produce distinct behavioral responses [106,107,108]. G protein and β-arrestin2 signaling through various mu agonists are mostly studied in the original mu opioid receptor, MOR-1. Various mu agonists can differentially induce receptor-β-arrestin interactions that block Receptor-G protein coupling and/or produce β-arrestin-dependent signaling. The hypothesis that G protein signaling produces analgesic responses while β-arrestin2 signaling is responsible for common side-effects has led to the effort to develop novel analgesic drugs that are G protein-biased and/or non-β-arrestin-biased [109,110]. Discovery of multiple OPRM1 7TM *C*-terminal variants raises questions about the roles of these variants in biased signaling via various mu opioids, including endogenous opioid peptides.

There are four arrestin subtypes encoded by four different genes: SAG, ARRB1, ARRB2 and ARR3. The SAG was isolated as Arrestin1 or a visual arrestin. The ARR3 was cloned as Arrestin4 or X-arrestin or a cone arrestin. The ARRB1 (Arrestin1) and ARRB2 (Arrestin2) were identified as non-visual arrestins, and also named as Arrestin2 and Arrestin3, respectively, which often cause confusion about their gene or protein identity in literature. Both Arrestin1 (ARRB1) and Arrestin2 (ARRB2) have been widely studied in GPCR field. Here we refer to arrestin2 as the gene product of the ARRB2, which was sometimes called arrestin3 in literature.

β-arrestin signaling is determined by β-arrestin recruitment assays. Several β-arrestin recruitment assays, such as the PathHunter (DiscoverX) [111], PRESTO-Tango assay [112], bioluminescen energy transfer (BRET) assay [113,114], Transfluor imaging assay [115] and NanoLuc Binary Technology [116], have been developed. Mu agonist-induced β-arrestin2 signaling in the 7TM *C*-terminal variants has been measured by the PathHunter assay in CHO cells stably expressing the EA-tagged β-arrestin2 and PK-tagged individual 7TM *C*-terminal variant. When β-arrestin2-EA and the 7TM variant-PK is expressed separately, there is no β-galactosidase activity. Yet, the physical interaction of β-arrestin2-EA with 7TM variant-PK induced by mu agonists reconstitutes the β-galactosidase activity that produces chemiluminescent signal in the presence of its substrate, which can be detected through a luminescent microplate reader.

Several endogenous opioid peptides, including β-endorphin, dynorphin, dynorphin A, [Met]^5^Enkephalin-Arg^6^-Phe^7^, and endomorphins 1/2, as well as several mu opioids such as DAMGO, morphine, fentanyl, buprenorphine and methadone, were used to investigate their abilities to induce β-arrestin2 recruitment on the mouse 7TM *C*-terminal variants. The results from concentration-response curves unveiled obvious differences in both potency (EC_50_ values) and efficacy (% Maximum effect, % E_max_) of the endogenous opioid peptides and mu opioids among the 7TM *C*-terminal variants (Table 4) [81]. For example, the EC_50_ values of [Met]^5^Enkephalin-Arg^6^-Phe^7^ and endomorphin-1 had over a 5-fold difference between mMOR-1O and mMOR-1E. [Met]^5^Enkephalin-Arg^6^-Phe^7^ was more potent in mMOR-1 than mMOR-1E. Similarly, β-endorphin was 5-fold more potent in mMOR-1O than in mMOR-1C.

The efficacy of the endogenous opioid peptides also varied among the mouse 7TM *C*-terminal variants. For example, [Met]^5^Enkephalin-Arg^6^-Phe^7^ was fully efficacious against mMOR-1C, but partially efficacious against mMOR-1 or mMOR-1O. Interestingly, [Met]^5^Enkephalin-Arg^6^-Phe^7^, endomorphin-1, and endomorphin-2 were more efficacious, but less potent, in mMOR-1A than in mMOR-1. The *C*-terminal tail of mMOR-1A contains four amino acids (aa) as VCAF, encoded by exon 3a, instead of the 12 aa, LENLEAETAPLP, encoded by exon 4 in mMOR-1. These results suggest that the *C*-terminal sequences can differentially influence the efficacy and potency in β-arrestin2 recruitment by these endogenous opioid peptides. No correlation between the EC_50_ and E_max_ values was observed. Like the endogenous opioid peptides, mu opioids such as morphine, fentanyl, and methadone also revealed marked differences in both potency and efficacy of β-arrestin2 recruitment among the mouse 7TM variants.

To compare β-arrestin2 recruitment with G protein coupling, [^35^S]GTPγS binding was performed in the same CHO cells used in the β-arrestin2 recruitment assay [81]. Again, the endogenous opioid peptides and mu opioids displayed differential profiles of [^35^S]GTPγS binding among the mouse 7TM variants [81]. Consequently, the bias factor can be mathematically determined by using the parameters from β-arrestin2 recruitment and [^35^S]GTPγS binding assays with the operational model of Black and Leff [117,118], a model commonly used in GPCR field, to see if an agonist is β-arrestin2-biased or G protein-biased. Heatmaps from the calculation revealed a wide range of differences in bias factors of the endogenous opioid peptides and mu opioids (Figure 4) [80]. When the bias factors were normalized to DAMGO at mMOR-1 (Figure 4A), [Met]^5^Enkephalin-Arg^6^-Phe^7^ showed the most G protein bias toward mMOR-1E, as indicated by the highest positive number (+24.5), while it was β-arrestin2-biased against mMOR-1O (-2). Similar scenarios were seen in endomorphin-1 and β-endorphin. When the bias factors of individual agonists were normalized to mMOR-1 (Figure 4B), all endogenous opioid peptides and mu opioids excluding endomorphin-2 clearly displayed β-arrestin2 bias toward mMOR-1O, an exon 7-associated 7TM variant, compared to mMOR-1. Similarly, all endogenous opioid peptides and mu opioids except for [Met]^5^Enkephalin-Arg^6^-Phe^7^ showed greater β-arrestin2 bias in mMOR-1B1. Interestingly, [Met]^5^Enkephalin-Arg^6^-Phe^7^ exhibited G protein bias toward all 7TM variants with the exception of mMOR-1O. These results underline the functional importance of these 7TM *C*-terminal variants on biased signaling induced by not only various mu opioids but also by endogenous opioid peptides.

Why do the *C*-terminal sequences have marked impact on biased signaling by endogenous opioid peptides and mu opioids in terms of G protein coupling and β-arrestin2 recruitment? One possible mechanism is that different *C*-terminal sequences contain various potential phosphorylation sites and differential phosphorylation induced by mu agonists can modulate G protein and/or β-arrestin2 signaling. The *C*-terminal tails encoded by exon 7 have a consensus phosphorylation code, PxPxxE/D or PxxPxxE/D, for high affinity arrestin binding that was predicted from the crystal studies of GPCRs [72]. When this code was mutated, mMOR-1O, an exon 7-associated 7TM variant was unable to recruit β-arrestin2 by mu agonists (unpublished data). This may explain why mMOR-1O had most β-arrestin2 bias toward most mu agonists including endogenous opioid peptides. Another possibility is that the *C*-terminal sequences can interact with intracellular loops of the receptor that are important for G protein or β-arrestin2 signaling or with other receptor-associated signaling proteins, a similar mechanism for the differences in the binding affinity of the endogenous opioid peptides among 7TM variants as mentioned above. Finally, different *C*-terminal tails may modulate receptor conformations favoring either G protein coupling or β-arrestin2 recruitment particularly induced by endogenous opioid peptides. Biased signaling has been referred to different signaling pathways produced by various agonists on a single GPCR. The results from the 7TM *C*-terminal variants offer another meaning of biased signaling in which a single agonist can stimulate divergent signaling pathways via multiple 7TM *C*-terminal variants.

## 7. Conclusions

Extensive alternative splicing of the OPRM1 gene creates multiple splice variants or receptor isoforms that are conserved from rodent to human, providing new insights into our understanding of the complex actions of various mu agonists, including endogenous opioid peptides. Like most mu opioids such as morphine and fentanyl, endogenous opioid peptides can differentially induce G protein coupling, β-arrestin2 recruitment, and biased signaling through various 7TM *C*-terminal splice variants. Variable binding affinities of endogenous opioid peptides toward the 7TM *C*-terminal variants indicate the influence of *C*-terminal tail sequences on overall receptor structure and/or ligand binding pockets for the endogenous opioid peptides. Future structural determination of such influences by the *C*-terminal sequences using new technologies such as high-resolution cryogenic electron microscopy would greatly advance our knowledge on the role of the 7TM *C*-terminal variants, especially in the pharmacology of endogenous opioid peptides. Although all the results presented in this review were obtained from in vitro cell models, they suggest the functional relevance of these 7TM *C*-terminal variants in mediating the actions of endogenous opioid peptides and mu opioids in vivo where they are co-expressed. The in vivo pharmacological function of an endogenous opioid peptide or a mu opioid should be considered as its combinational effects on different 7TM *C*-terminal variants. Region-specific or cell-specific expression of the 7TM *C*-terminal variants also raises questions on whether the 7TM *C*-terminal variants have distinct roles in a region-specific or cell-specific manner. It will be interesting to further investigate in vivo functions of each individual 7TM *C*-terminal variant using novel gene targeting animal models in which only one individual 7TM *C*-terminal variant is expressed.

## Figures and Tables

**Figure 1 ijms-22-03779-f001:**
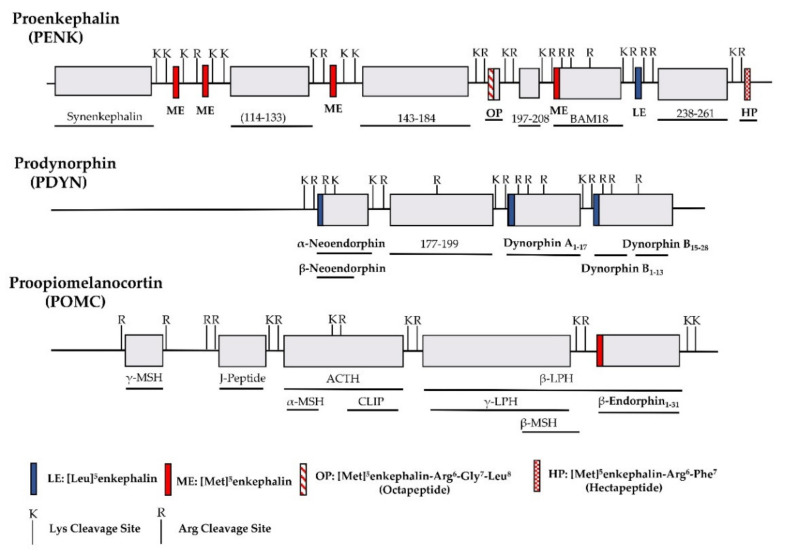
Schematic of the major endogenous peptides processed from human proenkephalin (PENK), prodynorphin (PDYN) and proopiomelanocortin (POMC). BAM: bovine adrenal medulla peptide; MSH: melanocyte stimulating hormone; ACTH: adrenocorticotropic hormone; CLIP: corticotropin-like intermediate lobe peptide; LPH: lipotropin.

**Figure 2 ijms-22-03779-f002:**
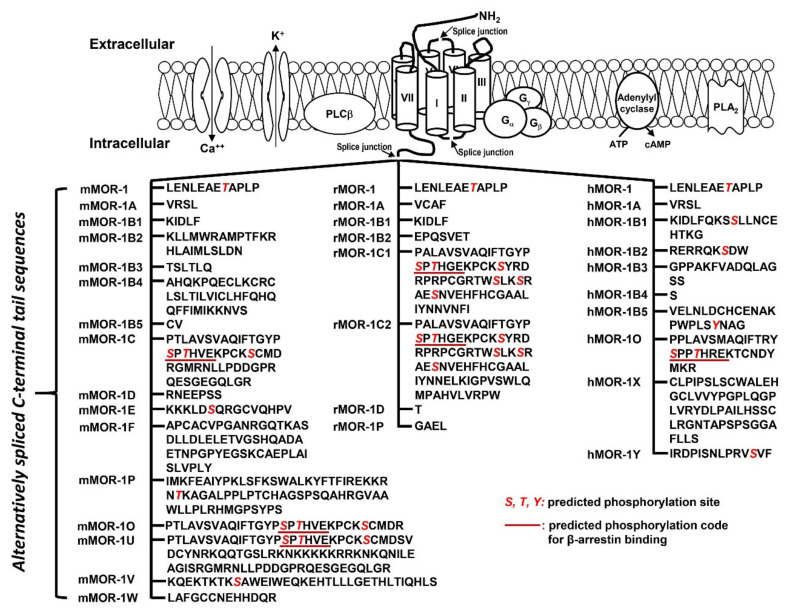
Predicted amino acid sequences from 7TM *C*-terminal variants (modified from [17]. The top panel is an animation that shows structures of MORs and adjacent proteins on membrane. TM domains are indicated by cylinders. Splice junctions are shown by arrows. Calcium (Ca++) and potassium (K+) channels are indicated by opened canals across membrane. Gα, Gβ and Gγ: G proteins; PLCβ: phospholipase Cβ; PLA2: phospholipase A2; The bottom panel listed predicted amino acid sequences encoded by downstream exons of exon 3 in mouse (mMOR), rat (rMOR) and human (hMOR) splice variants. Italic red S, T and Y are predicted phosphorylation sites. Underlined sequences are predicted phosphorylation codes, PxPxxE/D or PxxPxxE/D, for β-arrestin binding based on crystal G protein coupled receptors (GPCR) structures [72].

**Figure 3 ijms-22-03779-f003:**
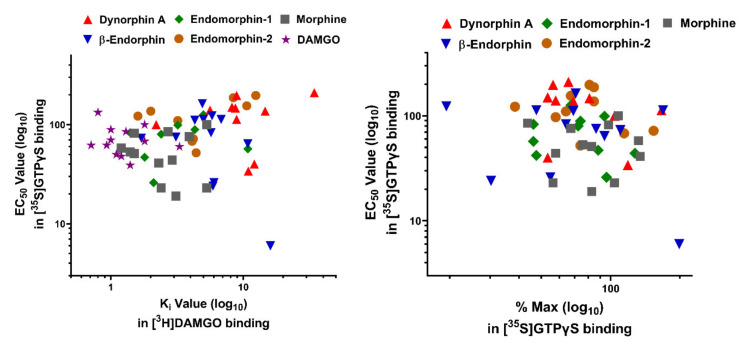
Correlation of the EC_50_ values with % maximum stimulation (% Max) in [^35^S]GTPγS binding and with the K_i_ values in receptor binding among mouse Oprm1 7TM *C*-terminal variants. A). Correlations of the K_i_ values in receptor binding from Table 2 with the EC_50_ values in [^35^S]GTPγS binding from Table 3. Correlation coefficients (*r^2^*) were calculated for each drug by linear regression (Prism 8, GraphPad). There was no significant correlation between binding site affinity (K_i_) and potency (EC_50_) in the [^35^S]GTPγS binding. DAMGO, *r^2^* = 0.03; Morphine, *r^2^* = 0.01; β-endorphin, *r^2^* = 0.24; Dynorphin A, *r^2^* = 0.16; Endomorphin-1, *r^2^* = 0.01; Endomorphin-2, *r^2^* = 0.44. B). Correlation of the EC_50_ values and % maximum stimulation (% Max) in the [^35^S]GTPγS binding. No significant correlation between the EC_50_ and % Max was observed. Morphine, *r^2^* = 0.00; β-endorphin, *r^2^* = 0.05; Dynorphin A, *r^2^* = 0.16; Endomorphin-1, *r^2^* = 0.04; Endomorphin-2, *r^2^* = 0.07.

**Figure 4 ijms-22-03779-f004:**
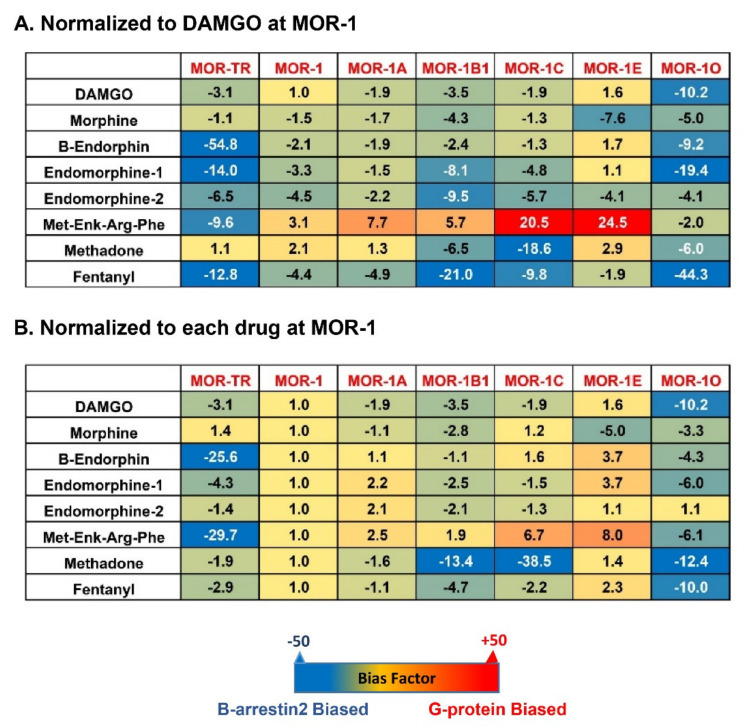
Heatmap of biased factors (adopted from [81]). Biased factors were calculated using the Black and Leff Operational Model by using different normalization methods, as described in [81]. (**A**). Normalized with respect to DAMGO at MOR−1 for a comparison between drugs and variants. (**B**). Normalized with respect to each drug at mMOR−1 for a comparison across variants. The negative (blue) values indicate β-arrestin2 bias whereas the positive bias (red) values indicate G protein bias.

**Table 1 ijms-22-03779-t001:** Amino acid sequences of selected human endogenous opioid peptides.

Precursor	Opioid Peptide	Copies of Peptide	Structure	Other Peptides
Proenkephalin (PENK)	[Leu]^5^enkephalin	1	Tyr-Gly-Gly-Phe-Leu	Synenkephalin
	[Met]^5^enkephalin	4	Tyr-Gly-Gly-Phe-Met	
	[Met]^5^enkephalin-Arg^6^-Gly^7^-Leu^8^(Octapeptide)	1	Tyr-Gly-Gly-Phe-Met-Arg-Gly-Leu	
	[Met]^5^enkephalin-Arg^6^-Phe^7^ (Heptapeptide)	1	Tyr-Gly-Gly-Phe-Met-Arg-Ph	
Prodynorphin (PDYN)	Dynorphin A_1-17_	1	Tyr-Gly-Gly-Phe-Leu-Arg-Arg-Ile-Arg-Pro-Lys-Leu-Lys-Trp-Asp-Asn-Gln	α-neoendorphin, β-neoendorphin, Big dynorphin, Leumorphin
	Dynorphin B_1-13_	1	Tyr-Gly-Gly-Phe-Leu-Arg-Arg-Gln-Phe-Lys-Val-Val-Thr	
Pro-opiomelanocortin (POMC)	β_h_-Endorphin_1-31_	1	Tyr-Gly-Gly-Phe-Met-Thr-Ser-Glu-Lys-Ser-Gln-Thr-Pro-Leu-Val-Thr-Leu-Phe-Lys-Asn-Ala-Ile-Ile-Lys-Asn-Ala-Tyr-Lys-Lys-Gly-Glu	γ-MSH, ACTH, α-MSH, CLIP, β-LPH, γ-LPH, β-MSH
Unknown	Endomorphin-1		Tyr-Pro-Trp-Phe-NH_2_	
	Endomorphin-2		Tyr-Pro-Phe-Phe-NH_2_	

MSH: melanocyte stimulating hormone; ACTH: adrenocorticotropic hormone; CLIP: corticotropin-like intermediate lobe peptide; LPH: lipotropin.

**Table 2 ijms-22-03779-t002:** Competition of [^3^H]DAMGO binding in Chinese Hamster Ovary (CHO) cells stably expressing mouse, rat and human Oprm1 7TM *C*-terminal variants.

	Ligand										
K_i_ Value (nM)	DAMGO	Morphine	Fentanyl	Methadone	M6G	β-Endorphin	Dynorphin A	Endomorphin 1	Endomorphin 2	[Met]^5^Enkephalin-Arg^6^-Phe^7^	Refs.
**Mouse**											
mMOR-1	1.8 ± 0.5	5.3 ± 2.0	2.3 ± 1.0	1.4 ± 0.1	5.2 ± 1.8	11 ± 2.9	11 ± 0.5	2.1 ± 0.8	4.2 ± 1.8	4.1 ± 1.0	[73,100]
mMOR-1A	1.0 ± 0.3	3.1 ± 0.5	1.5 ± 0.6	0.7 ± 0.1	5.0 ± 1.5	4.3 ± 1.0	8.2 ± 2.8			3.5 ± 1.3	[73,77]
mMOR-1C	0.93 ± 0.2	2.4 ± 0.6	1.2 ± 0.4	0.5 ± 0.1	4.1 ± 1.2	5.8 ± 0.5	5.6 ± 0.8	1.4 ± 0.4	1.6 ± 0.2	2.1 ± 0.7	[73,100]
mMOR-1D	0.71 ± 0.1	1.5 ± 0.2	3.3 ± 1.5	1.4 ± 0.1	4.8 ± 0.8	1.7 ± 0.4	2.2 ± 0.8	1.8 ± 0.3	2.0 ± 0.3	3.7 ± 1.2	[73,100]
mMOR-1E	1.2 ± 0.5	2.3 ± 0.4	1.2 ± 0.5	0.7 ± 0.3	5.6 ± 0.7	5.0 ± 1.2	8.9 ± 1.1	2.4 ± 0.1	4.4 ± 0.8	4.4 ± 0.9	[73,100]
mMOR-1B1	1.4 ± 0.2	5.3 ± 1.0			10 ± 1.6	6.8 ± 3.2	15 ± 7.1	11 ± 5.6	12 ± 1.5		[75]
mMOR-1B2	1.3 ± 0.1	3.9 ± 0.4			8.4 ± 1.3	4.9 ± 1.7	34 ± 18	5.0 ± 1.8	8.4 ± 1.1		[75]
mMOR-1B3	1.8 ± 0.9	1.5 ± 0.5			3.9 ± 1.3	3.1 ± 1.4	8.7 ± 1.8	3.2 ± 0.6	3.2 ± 0.8		[75]
mMOR-1B5	1.0 ± 0.3	1.4 ± 0.6			5.2 ± 0.1	5.7 ± 1.2	8.9 ± 2.3	4.3 ± 0.8	11 ± 1.8		[75]
mMOR-1F	1.1 ± 0.2	2.9 ± 0.5	1.7 ± 0.5	1.3 ± 0.2	9.6 ± 0.8	6.0 ± 1.6	12 ± 1.0	2.9 ± 0.5	4.1 ± 1.3	3.9 ± 0.8	[73,78]
mMOR-1O	3.3 ± 1.2	2.7 ± 0.6			17 ± 1.0	16 ± 5.3	58 ± 26				[77]
mMOR-1P	0.8 ± 0.3	1.2 ± 0.8			11 ± 3.4	5.9 ± 2.4	103 ± 23				[77]
**Rat**											
rMOR-1	3.3 ± 0.6	5.6 ± 0.8			17 ± 2.2	3.7 ± 0.4	12 ± 3.0	4.1 ± 0.7	8.0 ± 2.0		[74]
rMOR-1A	6.0 ± 0.9	8.0 ± 0.4			26 ± 2.1	11 ± 0.6	23 ± 1.6	6.5 ± 0.3	12 ± 0.6		[74]
rMOR-1C1	4.5 ± 0.9	7.4 ± 0.3			25 ± 2.4	8.8 ± 0.5	13 ± 2.3	3.9 ± 0.1	10 ± 0.6		[74]
rMOR-1D	4.7 ± 1.2	7.4 ± 0.5			21 ± 1.8	8.5 ± 0.6	11 ± 1.7	3.9 ± 0.4	7.5 ± 0.4		[74]
**Human**											
hMOR-1	1.2 ± 0.2	2.2 ± 0.9			10 ± 0.3	15 ± 11.0	87 ± 14	4.2 ± 1.4	15 ± 7.1		[76]
hMOR-1B1	1.2 ± 0.4	2.4 ± 1.1			5.0 ± 0.2	7.8 ± 1.5	19 ± 6.6	3.8 ± 0.8	5.4 ± 0.6		[76]
hMOR-1B2	5.2 ± 1.4	11 ± 3.5			42 ± 7.9	25 ± 5.1	49 ± 22	12 ± 0.1	20 ± 1.3		[76]
hMOR-1B3	1.8 ± 0.5	3.2 ± 0.6			16 ± 1.2	8.2 ± 2.2	14 ± 2.3	4.9 ± 1.5	6.3 ± 1.5		[76]
hMOR-1B4	2.3 ± 0.6	5.5 ± 1.7			23 ± 7.4	16 ± 0.4	71 ± 30	9.9 ± 2.3	23 ± 2.0		[76]
hMOR-1B5	2.1 ± 0.4	3.9 ± 0.9			12 ± 2.6	10 ± 3.4	53 ± 23	4.6 ± 0.3	9.6 ± 3.0		[76]
hMOR-1O	2.2 ± 0.6	2.0 ± 0.7			16 ± 2.6		25 ± 8.5				[79]
hMOR-1X	2.1 ± 0.2	2.7 ± 1.0			17 ± 5.3		187 ± 27				[79]
hMOR-1Y	2.5 ± 0.8	4.3 ± 1.7			8.3 ± 2.2	8.4 ± 1.8	25 ± 13	5.1 ± 1.1	9.4 ± 3.0		[76]

[^3^H]DAMGO binding was performed with membranes prepared from CHO cells stably expressing indicated splice variants, as described in indicated references.

**Table 3 ijms-22-03779-t003:** Mu agonist-induced [^35^S]GTPγS binding in Chinese Hamster Ovary (CHO) cells stably expressing mouse, rat and human mu opioid receptor gene (Oprm1) 7TM *C*-terminal variants.

	Ligand														
	DAMGO		Morphine		β-Endorphin		Dynorphin A		Endomorphin 1		Endomorphin 2		[Met]^5^Enkephalin-Arg^6^-Phe^7^		Ref.
	*EC_50_* (nM)	*%Max*	*EC_50_* (nM)	*%Max*	*EC_50_* (nM)	*%Max*	*EC_50_* (nM)	*%Max*	*EC_50_* (nM)	*%Max*	*EC_50_* (nM)	*%Max*	*EC_50_* (nM)	*%Max*	
**Mouse**															
mMOR-1	68 ± 4	100	23 ± 2	102 ± 5	64 ± 7	97 ± 2	34 ± 9	109 ± 7	26 ± 4	98 ± 8	72 ± 11	124 ± 8	53 ± 3	118 ± 15	[73]
mMOR-1A	70 ± 3	100	19 ± 4	91 ± 2	111 ± 27	83 ± 3	150 ± 36	73 ± 6	42 ± 13	69 ± 2	97 ± 28	76 ± 3	133 ± 9	75 ± 4	[73]
mMOR-1C	62 ± 4	100	23 ± 5	75 ± 4	123 ± 19	44 ± 3	140 ± 19	76 ± 10	83 ± 20	68 ± 15	122 ± 46	62 ± 15	60 ± 17	51 ± 2	[73]
mMOR-1D	62 ± 6	100	82 ± 34	99 ± 3	73 ± 18	105 ± 6	100 ± 41	102 ± 6	47 ± 21	94 ± 8	137 ± 24	92 ± 5	170 ± 16	94 ± 3	[73]
mMOR-1E	48 ± 4	100	41 ± 13	116 ± 4	113 ± 25	130 ± 3	113 ± 9	129 ± 9	80 ± 4	85 ± 9	52 ± 26	86 ± 8	131 ± 19	94 ± 10	[73]
mMOR-1B1	39 ± 8	100	100 ± 38	104 ± 38	113 ± 47	69 ± 21	137 ± 69	83 ± 23	57 ± 23	68 ± 19	197 ± 95	90 ± 0			[75]
mMOR-1B2	85 ± 18	100	76 ± 13	82 ± 8	163 ± 22	84 ± 5	210 ± 25	81 ± 6	126 ± 29	82 ± 8	187 ± 23	92 ± 4			[75]
mMOR-1B3	100 ± 14	100	51 ± 6	91 ± 3	75 ± 19	93 ± 2	147 ± 56	90 ± 6	99 ± 1	97 ± 2	110 ± 6	80 ± 3			[75]
mMOR-1B5	89 ± 13	100	53 ± 4	87 ± 7	83 ± 27	80 ± 4	197 ± 32	75 ± 3	89 ± 13	86 ± 7	155 ± 8	82 ± 4			[75]
mMOR-1F	50 ± 6	100	44 ± 17	76 ± 13	26 ± 6	74 ± 7	40 ± 8	73 ± 3	44 ± 18	113 ± 5	68 ± 18	107 ± 4	29 ± 9	94 ± 16	[73]
mMOR-1O	60 ± 19	100	85 ± 31	66 ± 23	6 ± 1	141 ± 8									[77]
mMOR-1P	133 ± 23	100	58 ± 9	115 ± 23	24 ± 5	55 ± 3									[77]
**Rat**															
rMOR-1	12 ± 3	100			4 ± 2	105.58			14 ± 4	137.34					[74]
rMOR-1A	13 ± 5	100			13 ± 5	100.57			15 ± 3	116.48					[74]
rMOR-1C1	74 ± 22	100			48 ± 4	154.94			54 ± 8	161.80					[74]
rMOR-1D	125 ± 26	100			91 ± 14	146.02			100 ± 26	128.32					[74]
**Human**															
hMOR-1	120 ± 17	100	21 ± 4	97.57	4 ± 1	68.75	296 ± 16	36.46							[76]
hMOR-1A	161 ± 21	100	30 ± 2	121.31	8 ± 2	71.31	36 ± 1	63.93							[76]
hMOR-1B1	255 ± 46	100	41 ± 5	64.41	25 ± 6	57.97	63 ± 17	50.51							[76]
hMOR-1B2	1028 ± 68	100	77 ± 9	80.00	73 ± 10	97.84	292 ± 66	97.84							[76]
hMOR-1B3	549 ± 86	100	86 ± 19	65.44	33 ± 11	61.78	98 ± 27	39.38							[76]
hMOR-1B4	341 ± 65	100	38 ± 5	71.68	19 ± 2	65.32	58 ± 14	40.75							[76]
hMOR-1B5	936 ± 233	100	90 ± 18	61.46	55 ± 2	92.01	158 ± 15	81.60							[76]
hMOR-1Y	571 ± 255	100	100 ± 20	88.05	43 ± 3	73.18	100 ± 21	77.26							[76]

[^35^S]GTPγS binding assay was performed with membranes prepared from CHO cells stably expressing indicated splice variants, as described in indicated references. The percentage of maximum stimulation (% Max) of the agonists was normalized with that of DAMGO.

**Table 4 ijms-22-03779-t004:** EC_50_/K_i_ value ratios of mu agonists among the mouse, rat, and human mu opioid receptor gene (OPRM1) 7TM *C*-terminal variants.

	Ligand							
	DAMGO	Morphine	β-Endorphin	Dynorphin A	Endomorphin-1	Endomorphin-2	[Met]^5^Enkephalin-Arg^6^-Phe^7^	Refs.
	EC_50_/K_i_	EC_50_/K_i_	EC_50_/K_i_	EC_50_/K_i_	EC_50_/K_i_	EC_50_/K_i_	EC_50_/K_i_	
**Mouse**								
mMOR-1	38	4	6	3	12	17	13	[73,100]
mMOR-1A	70	6	26	18			38	[73,77]
mMOR-1C	67	10	21	25	59	76	29	[73,100]
mMOR-1D	87	55	43	45	26	69	46	[73,100]
mMOR-1E	40	18	23	13	33	12	30	[73,100]
mMOR-1B1	28	19	17	9	5	16		[75]
mMOR-1B2	65	19	33	6	25	22		[75]
mMOR-1B3	56	34	24	17	31	34		[75]
mMOR-1B5	89	38	15	22	21	15		[75]
mMOR-1F	45	15	4	3	15	17	7	[73,78]
mMOR-1O	18	31	0.4					[77]
mMOR-1P	166	48	4					[77]
**Rat**								
rMOR-1	4		1		3			[74]
rMOR-1A	2		1		2			[74]
rMOR-1C1	16		5		14			[74]
rMOR-1D	27		11		26			[74]
**Human**								
hMOR-1	100	10	0.3	3				[76,79]
hMOR-1B1	213	17	3	3				[76]
hMOR-1B2	198	7	3	6				[76]
hMOR-1B3	305	27	4	7				[76]
hMOR-1B4	148	7	1	0.8				[76]
hMOR-1B5	4	23	6	3				[76]
hMOR-1Y	228	23	5	4				[76]

## Data Availability

Not applicable.

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
