# Peer review of "Endogenous Opioid Peptides and Alternatively Spliced Mu Opioid Receptor Seven Transmembrane Carboxyl-Terminal Variants"

_ijms, 2021, doi:10.3390/ijms22073779_

Round 1

Reviewer 1 Report

Manuscript ID: ijms-1172253

Type of manuscript: Review

Title: Endogenous Opioid Peptides and Alternatively Spliced Mu Opioid

Receptor Seven Transmembrane Carboxyl-Terminal Variants

Authors: Anna Abrimian, Tamar Kraft, Ying-Xian Pan

Pan et al. review focus on mechanisms and functions of endogenous opioid peptides mediated through the OPRM1 7TM C-terminal splice variants. To this aim authors collected and summarized, critically commenting, results from early receptor binding affinity, G protein activation, β-arrestin2 recruitment and biased signaling. The review is well written and it is easily deducible the take-home message. Building upon that this review could provide new insights into the mechanisms and functions of endogenous opioid peptides, mediated through the OPRM1 7TM C-terminal splice variants, it is suggested its publication after minor revisions.

  1. Page 1 line 12. Remove the adjective endogenous, it is redundant.
  2. Page 1 line. What do authors mean with “mu opioid”? Ligands? Agonist or antagonist compounds? Peptidic or non-peptidic compounds?
  3. Page 1 line 34. As point 1.
  4. Page 1 lines 35-38. Simplify the period to increase readability.
  5. Page 2 line 51 specify the TM C- abbreviation.
  6. Page 2 line 54-58. Remove the entire period because it isn’t pertinent with thwi aims introduced by author considering that it is well known that nociceptin isn’t an opioid peptide.
  7. Page 2 line 65. Replace delta opioid receptor with abbreviation.
  8. Page 2 line 67. Do you mean CNS?
  9. Page 2 lines 78-81. Re-write the sentences they are low reading.
  • Page 2 line 88 Italicize N- and C- (as well as in the whole text).
  • Figure 1. Insert the legend in the figure e remove it from the caption.
  • Page 4 line 109 correct central nerve system in central nervous system CNS.
  • Page 4 line 109 replace system with functions.
  • Page 5 lines 121-123. Correct the sentence. Heroin isn’t a clinically used MOR analgesic!!!
  • Page 5 line 176 correct morphine-induce with morphine-induced or morphine-inducing.
  • Page 8 line 273 remove full stop.
  • Page 9 line 315 correct radio with ratio (as well as in the following lines).
  • Regarding the biased signalling or functional selectivity it is suggested to cite more updated references (i.e. Turnaturi et al. 2019).
  • Page 11 line 402 explicit aa abbreviation.
  • Uniform in the whole main text mu in MOR.

Reviewer 2 Report

In the submitted review article, the authors describe the 7TM  μ opioid receptor splice variants and their in vitro profiles with their opinions. The contents would be informative for many readers. Especially, the heatmap of biased factors seems to be very interesting. So, the submitted manuscript would be acceptable as a review article in this journal. However, the authors should consider the following points before the publication.

  1. The term “configuration” is incorrectly used. The configurations mean tree-dimensional structures of the molecules, that is stereochemistries of the molecules. So, changing the configuration of a molecule always means that bonds (covalent bonds) are broken, and a different configuration is a different molecule. On the other hand, conformations mean the temporary molecular shapes that results from such a bond-rotation. So, conformations of a molecule are readily interconvertible, and are all the same molecules. The interaction between ligands and receptors can never change the stereochemistry of ligands and receptors (any covalent bonds in ligands and receptors never be broken). It is the conformation that is changed when ligands bind to their target receptors. Therefore, “configuration” should be changed to “conformation.”

  1. The authors described that although dynorphins are considered as kappa1-opioid receptor (KOR-1), they bind to mu-opioid receptor (MOR-1) and DOR-1 with high affinities. Also, β-endorphin has high affinity for MOR-1 and DOR-1 (lines 90-92). In general, dynorphins and β-endorphin are considered endogenous agonists for the κ and μ opioid receptors, respectively. Therefore, the appropriate references should be sited.

  1. The authors described that …by most mu agonists such as DAMGO, morphine, fentanyl, M6G and methadone, in contract to their unchanged binding affinity (Table 2) (lines 333-334). However, Table 2 does not show any data of fentanyl, M6G and methadone.

  1. As the manuscript deals with the results obtained by in vitro experiments, not dose-response curves (lines 266 and 390) but concentration-response curves are suitable.

  1. Minor points: β-endorphin is not correctly sowed in Table 4. “B” not "β."
